# Physiological and Transcriptome Responses of Sweet Potato [*Ipomoea batatas* (L.) Lam] to Weak-Light Stress

**DOI:** 10.3390/plants13162214

**Published:** 2024-08-09

**Authors:** Jin Yang, Huanhuan Qiao, Chao Wu, Hong Huang, Claude Nzambimana, Cheng Jiang, Jichun Wang, Daobin Tang, Weiran Zhong, Kang Du, Kai Zhang, Changwen Lyu

**Affiliations:** 1College of Agronomy and Biotechnology, Southwest University, Chongqing 400715, China; 15178857808@163.com (J.Y.);; 2Special Crops Institute, Chongqing Academy of Agricultural Sciences, Chongqing 402160, China; 3Key Laboratory of Biology and Genetic Breeding for Tuber and Root Crops in Chongqing, Beibei, Chongqing 400715, China; 4Engineering Research Center of South Upland Agriculture, Ministry of Education, Chongqing 400715, China; 5Human Resources Department, Southwest University, Chongqing 400715, China

**Keywords:** *Ipomoea batatas* (L.) Lam, weak light, agronomic traits, photosynthetic physiology, transcriptome analysis

## Abstract

In the relay intercropping system of maize/sweet potato, the growth of the sweet potatoes is seriously limited by weak light stress in the early stage due to shade from maize plants. However, it is not clear how the weak light affects sweet potatoes and causes tuberous root loss. By setting two light intensity levels (weak light = 30% transmittance of normal light), this study evaluated the responses of two sweet potato cultivars with different tolerances to weak light in a field-based experiment and examined the divergence of gene expression related to light and photosynthesis in a pot-based experiment. The results showed that under weak light, the anatomic structure of functional leaves changed, and the leaf thickness decreased by 39.98% and 17.32% for Yuhongxinshu-4 and Wanshu-7, respectively. The ratio of S/R increased, and root length, root superficial area, and root volume all decreased. The photosynthetic enzyme rubisco was weakened, and the net photosynthetic rate (Pn) declined as well. The level of gene expression in Wanshu-7 was higher than that of Yuhongxinshu-4. The KEGG analysis showed that differentially expressed genes from the two cultivars under weak-light stress used the same enrichment pathway, mainly via glutathione metabolism and flavonoid biosynthesis. After full light levels were restored, the differentially expressed genes were all enriched in pathways such as photosynthesis, photosynthetic pigment synthesis, and carbon metabolism. These findings indicated that weak light changed the plant morphology, photosynthetic physiology and gene expression levels of sweet potatoes, which eventually caused losses in the tuberous root yield. The more light-sensitive cultivar (Wanshu-7) had stronger reactions to weak light. This study provides a theoretical basis and strategy for breeding low-light-tolerant varieties and improving relay intercropping production in sweet potatoes.

## 1. Introduction

The shortage of land resources is an urgent problem in China. Intercropping planting has become one of the important measures to improve land resource utilization efficiency and crop yield [1,2,3]. The sweet potato [*Ipomoea batatas* (L.) Lam.] is one of China’s main crops. Due to its high yield potential and low input requirement, it is widely planted in the mountainous areas of southwestern China, such as Chongqing and Sichuan and Guizhou Province. In Chongqing and Sichuan Province, there are more than 900,000 hectares of sweet potato, and about half the crops are planted through relay intercropping with maize. This planting method can make good use of agricultural sources such as light, heat, water and land and enhance the multiple-cropping index. However, the collocation of maize and sweet potato leads to a shady environment for sweet potato in the symbiotic period, resulting in weak light, a low ratio of red light to far-red light, and inferior photosynthesis [4,5,6]. During the symbiotic period of sweet potato and maize in this planting mode, the photosynthetic effective radiation caused by corn shading is about 300 μmol m^−2^s^−1^ and is only 30–40% of that under normal light conditions. This light intensity is far below the light saturation point of sweet potato, which is usually more than 600 μmol m^−2^s^−1^ [7,8]. This seriously affects the morphogenesis and formation and expansion of storage roots, ultimately reduction in the sweet potato yield of about 40–50%. Inadequate sunlight severely influences the photosynthetic process in plants and can go on to cause significant changes in plants’ morphological characteristics, and trigger physiological and biochemical reactions [9,10,11].

These changes in crop morphological traits suggest that weak-light stress affects the normal growth and development of crops from three aspects: source, sink and flow. In crops, weak-light stress results in significant changes in plant height, stem diameter, leaf area, root length, and root volume [12,13,14]. Plants respond to low-light stress by expanding leaf area and increasing carbohydrate reserves [15]. For example, strawberry plants’ adaptation to short-term low-light stress was studied in [16]. After 10 days of treatment, the plants’ stem diameter, height, leaf area, leaf number, and petiole length were significantly inhibited. Under low-light stress, plants not only rebuild their morphological traits but also change their dry matter accumulation and distribution. Studies on tomatoes [17], rice [18], maize [13], and other crops have shown that low-light stress seriously affects the accumulation, transformation, and distribution of dry matter. Because different varieties of crops adapt to weak-light stress differently, their yields also differ [19]. In a study examining the mechanism by which rice adapts to low-light stress, it was found that shading reduced the number of tillers, effective tillers, and grains per panicle, as well as dry matter accumulation and 1000-grain weight, thus affecting the overall yield [20].

Functional leaves are highly sensitive to environmental changes, but they are also flexible in their responses. As the main organ for photosynthesis, they display the most obvious responses to weak light, including changes to their anatomical structure and physiological pathways. These adaptable changes can optimize the plants’ growth and development [21], make full use of weak light, and reduce adverse impacts. The anatomical structure of leaves undergoes adaptive changes in response to various abiotic stresses and these changes reflect the influence of environmental factors on the growth of plant leaves [22,23]. Under weak light, the ratio of palisade tissue to spongy tissue (also known as the palisade sea ratio) decreases, but the decline rates differ among species depending on the shade tolerance of the different plants [24,25]. For instance, corn leaves adapt to low-light environments by adjusting their biochemical traits and chloroplast content [26].

Photosynthetic parameters indicate the changes in photosynthesis in plants. For example, mustard seeds adapt to weak light by means of decreased stomatal conductivity (Gs), transpiration rate (Tr), net photosynthetic rate (Pn), electron transfer rate (ETR), ΦPSⅡ, and chlorophyll fluorescence quenching [27]. The Pn and Gs of sweet potato leaves have been shown to decrease significantly as a result of shading; however, intercellular CO_2_ concentration (Ci) increases significantly, which suggests that weak light is disadvantageous to the growth of sweet potatoes [14,28]. Furthermore, shading stress significantly weakens the photosynthetic capacity of mung bean leaves, as a result of changes in their morphological characteristics and increased light absorption under low-light conditions [29]. These changes occur due to the destruction of the anatomical structure of the leaves and the palisade and sponge tissues.

Under weak light conditions, the plants’ photosynthetic pigment content increases up to a certain point. The longer a plant is under weak-light stress, the more the photosynthetic pigment content increases. Plants which are more tolerant of weak light show a more marked increase in photosynthetic pigment [14,30,31,32]. Rubisco is the key enzyme in photosynthesis; as such, its activity in weak light declines significantly, which reduces efficiency of carboxylation. Shading decreased the activity of photosynthesis-related enzymes and RuBP carboxylase in sweet potatoes [14]. In addition, the mechanism of plant stress tolerance can be seen at the transcriptional level through RNA-seq technology, and some key functional genes have been screened for breeding stress-resistant varieties in crops. For example, a study found that under salt stress, the salt tolerance of sweet potatoes was mainly affected by transcription factors, protein detoxification, and hormone-related genes, all of which help improve the salt tolerance of sweet potatoes [33]. The maintenance of grain yield from tolerant rice plants under low-light conditions may be the result of accelerated gene expression, which enables plants to maintain the same rate of photosynthesis even under low-light conditions [34]. Regarding cold stress, four key genes (*pyl*, *PP2C*, *SnRK2*, *ABF*) in the ABA signal transduction pathway were found to be closely related to the cold stress response.

However, few studies have been undertaken on weak-light stress in sweet potatoes. Our previous studies have screened three sweet potato varieties with different sensitivities to low light using a cluster analysis [14]. The physiological and biochemical processes and molecular mechanisms of sweet potato under low-light stress are still unclear. Therefore, the aim of the present study was to investigate the specific traits of sweet potato varieties that are tolerant to low-light stress, including differences in growth and development, leaf anatomical structure and photosynthetic characteristics, and enzyme activity, as well as transcriptome responses to weak light. These analyses provided new insights for breeding weak-light-tolerant varieties of sweet potato and improving the relay intercropping of sweet potato with other crops.

## 2. Results

### 2.1. Effects of Weak-Light Stress on the Growth and Development of Sweet Potato

During the shading period (45 days post-transplant, or 45 DPT), the stem diameter of the two varieties in weak light was significantly less compared to the control (Table 1). Even 15 days after natural light was restored (15 DPR), the stem diameters of the Wanshu-7 plants were still affected by the weak light they had experienced in the earlier stage of the experiment. The main vine length of Yuhongxinshu-4 increased significantly in a shaded environment, but that of Wanshu-7 changed little. At 15 DPR, the difference between the main vine lengths of plants exposed to weak light and the control was the same as that at 45 DPT. The aboveground dry matter weight of the two varieties was significantly lower under weak-light stress than in natural light. However, after removing the shade net, the aboveground dry matter increased rapidly and was approaching that of the control at 15 DPR. In addition, the changes in those traits for Wanshu-7 (the more light-sensitive cultivar) were more dramatic than for Yuhongxinshu-4.

Meanwhile, for both cultivars, the shoot/root (S/R) ratio under weak light conditions was higher than that of the control at 45 DPT, and the S/R had increased even more by 15 DPR, which illustrated that, after natural light was restored, the underground part grew faster and accumulated more dry matter. However, for the Wanshu-7 plants, the difference in S/R between weak light and control increased after natural light was restored.

Weak-light stress significantly reduced the total root length, root surface area, and root volume of two sweet potato varieties (Figure 1A–C), and in Yuhongxinshu-4 the average root diameter also decreased significantly (Figure 1D). This finding suggests that weak-light stress in the symbiotic period could induce aboveground parts of the plant to overgrow and seriously affect root development. In contrast, tuberous roots began to expand faster after natural light was restored.

The performance in terms of yield and quality is shown in Table 2. The number and yield of tuberous roots per plant of Yuhongxinshu-4 were significantly lower in plants exposed to weak light than the control, but there were no obvious differences in soluble protein content or soluble sugar content between the two. For Wanshu-7, there were significant differences in storage root yield and soluble sugar content in plants exposed to weak light compared to those which remained in natural light throughout. In addition, the yield of the two varieties decreased to varying degrees compared to the corresponding controls. The yield reduction rates of Yuhongxinshu-4 and Wanshu-7 were 35.51% and 41.84%, respectively. These findings further indicate that the number of sweet potato storage roots in light-sensitive cultivars may be a genetic characteristic and be less affected by light intensity, whereas yield and soluble sugar content may be susceptible to light levels.

### 2.2. Effects of Weak-Light Stress on the Anatomical Structure and Photosynthetic Physiological Characteristics of Sweet Potato

When exposed to weak light for 45 days continuously, the anatomical structure of the leaf changed, becoming much thinner compared to the control (Figure 2). Compared to the control, the leaf thickness decreased in weak light. It was 237.63 μm in natural light and 142.61 μm in weak light for Yuhongxinshu-4, while for Wanshu-7, it was 186.35 μm in natural light and 154.08 μm in weak light (Figure 3A); the thickness declined by 39.98% and 17.32%, respectively. The thickness of palisade tissue, the thickness of sponge tissue, and the ratio of palisade tissue to sponge tissue also decreased to varying degrees in both cultivars. At 45 DPT, the palisade tissue thickness of Yuhongxinshu-4 decreased significantly, but the sponge tissue thickness decreased only slightly. From Figure 3D, it can be seen that the ratio of palisade tissue to sponge tissue for both sweet potato varieties decreased greatly, mainly due to the reduction in palisade tissue. As the main photosynthetic site of the leaves, the sponge tissue was less affected. However, after natural light had been restored for 15 days, there were no significant differences between weak-light and control plants in the total thickness of the leaves, palisade tissue, sponge tissue, or their ratio. This suggests that the effect of weak-light stress on leaf structure was temporary and that leaf thickness could recover to normal status in continuous natural light.

For the two cultivars, at 45 DPT, the Pn decreased greatly, and the intercellular CO_2_ concentration (Ci) increased under weak light (Figure 4A,C), while the Gs and Tr changed little (Figure 4B,D). At 15 DPR, for the same variety, there were no longer any noticeable differences in Pn and Ci between weak light and control plants. The Pn of sweet potato previously exposed to weak light increased after natural light was recovered in both cultivars. These results suggest that the intensity of light has a significant impact on photosynthetic parameters, especially on Pn, which in turn affected the accumulation of photosynthetic products and sweet potato yield.

Furthermore, it appears that weak-light stress may increase the photosynthetic pigment content of functional leaves in sweet potatoes (Table 3). After 45 days of shading, the chlorophyll a and chlorophyll b content increased significantly in both sweet potato varieties compared to the corresponding controls. At the same time, the ratio of chlorophyll a to chlorophyll b decreased by varying degrees. Such a change would be conducive to the capture of light energy and facilitate photosynthesis under weak light. At 15 DPR, the content of photosynthetic pigments in sweet potato leaves showed no noticeable differences between plants exposed to weak light and control plants.

As a key enzyme related to photosynthesis, rubisco’s activity is greatly influenced by light intensity (Figure 5). In this experiment, rubisco activity decreased by 12.62% in Yuhongxinshu-4 and by 46.61% in Wanshu-7 at DPT when plants were exposed to weak light compared with the control. The range of decline for Wanshu-7 was greater than for Yuhongxinshu-4, which further suggests that Wanshu-7 is more sensitive to weak light. However, when natural light was restored, rubisco activity improved, and the difference between plants exposed to weak light and control plants disappeared. These findings show that weak light in the early stages of sweet potato growth reduces rubisco activity leading to less photosynthesis, which may cause losses in the yield of storage roots. However, the response of the two cultivars differed greatly.

### 2.3. Transcriptomic Analysis of Sweet Potato Leaves in Response to Weak-Light Stress

There were 24 samples for Yuhongxinshu-4 and Wanshu-7 in total (shown in Section 4.2. Experimental setup). High-throughput sequencing obtained 157.75 Gb of clean data through filtering, and each sample had over 5.77 Gb of clean data. The Q30 base levels did not go below 93.54%. After comparing the filtered reads using HISAT2, it was found that 71.66–77.22% of clean reads could be compared to the reference genome (Table 4). This indicated that the transcriptome sequencing results were reliable and could subsequently be used for bioinformatics analysis.

An analysis of the number of differentially expressed genes (Figure 6A) showed that compared to the control, Yuhongxinshu-4 had 1173 differentially expressed genes at 30 DPT and 245 differentially expressed genes at 15 DPR. Similarly, compared with the control, Wanshu-7 had 7036 differentially expressed genes at 30 DPT and 1453 at 15 DPR. The number of differentially expressed genes in Yuhongxinshu-4 and Wanshu-7 decreased rapidly after light restoration, indicating that weak-light stress causes changes in the expression of a large number of genes, but that after natural light is restored, the expression levels of some genes gradually recover. The transcriptome changes in Wanshu-7 were stronger than in Yuhongxinshu-4 under weak-light stress, which indicates that Wanshu-7 was more sensitive to weak-light stress.

By means of a Venn plot, transcriptome differential analysis was conducted on the eight treatments, and 9425 genes were differentially expressed in total (Figure 6B). At 30 DPT, there were 151 differentially expressed genes co-expressed between Yuhongxinshu-4 and Wanshu-7, and at 15 DPR, 51 differentially expressed genes were co-expressed between the two varieties. The differentially expressed genes co-expressed by the different varieties may be the key genes in response to weak-light stress in sweet potatoes.

The results of a GO enrichment analysis of differentially expressed genes in the leaves of two sweet potato varieties in different light stress are shown in Figure 7. During the shading period, fifteen differentially expressed genes were involved in the response to light stimuli for Yuhongxinshu-4, five differentially expressed genes were involved in the response to UV light, and seven differentially expressed genes were enriched in the cellular response to environmental stimuli (Figure 7A). Fifty-one differentially expressed genes with molecular functions were significantly enriched in the Oxidoreductase activity, and 27 genes were significantly enriched in the transmembrane transporter activity (Figure 7B). For Wanshu-7, 11 differentially expressed genes were significantly enriched in response to light intensity and 14 genes were significantly enriched in the glutathione metabolic process (Figure 7C). In term of molecular function, differentially expressed genes were significantly enriched in pathways such as the structural constitution of ribosomes, ADP binding, oxidoreductase activity, acting on paid donors, with the incorporation or reduction in molecular oxygen, and glutathione transfer activity (Figure 7D).

However, after restoring natural light for 15 days (15 DPR), The differentially expressed genes of Yuhongxinshu-4 were significantly enriched in biological processes, such as the fructose 6-phosphate metabolic process, sucrose biosynthetic process, photosynthesis, and response to oxidative stress pathways. Most of the genes enriched in the GO pathway were upregulated genes (Figure 7E). In molecular function, differentially expressed upregulated genes were significantly enriched in fructose 1,6-bisphosphonate 1-photosphase activity, chlorophyll binding, and electron transport, transferring electrons within the cytochrome b6/f complex of photosystem II activity pathways (Figure 7F). In Wanshu-7’s biological processes, pathways such as photosynthesis, chlorophyll biosynthetic process, photosystem II stabilization, and photosynthetic electron transport in photosystem I were greatly enriched. However, most of them were downregulated genes (Figure 7G). In terms of molecular function, 11 downregulated genes were enriched in oxidoreductase activity, acting on NAD (P) H, quinone, or a similar compound as an acceptor. Forty-two differentially expressed genes were enriched in monooxygenase activity. Seven genes were enriched in the electron transporter and two were enrich in the pathway involved in transferring electrons within the cytochrome b6/f complex of photosystem II activity (Figure 7H).

Some differentially expressed genes related to weak-light stress were acquired from those differentially expressed genes related to photosynthesis, according to the GO enrichment analysis (Table 5). At 30 DPT, multiple genes involved in oxidoreductase were significantly downregulated. For example, *Tai6.44472* and *Tai6.31070* were significantly downregulated in Yuhongxinshu-4 and Wanshu-7. *Tai6.18802*, which is involved in photosystem II in photosynthesis, and *Tai6.21350*, which is related to ferriredoxin, were also significantly downregulated in Yuhongxinshu-4. Multiple genes involved in carbon fixation were differentially expressed in the two sweet potato varieties. For instance, *Tai6.12471* was significantly downregulated in Yuhongxinshu-4, and *Tai6.15090* was significantly upregulated in Wanshu-7. But, at 15 DPR, photosystem II 22 kDa protein (*Tai6.18802*, *Tai6.19673*, *Tai6.28593*) was upregulated in Yuhongxinshu-4, while a few genes involved in photosynthesis and carbon fixation—such as *Tai6.21350*, *Tai6.27478*, and *Tai6.35389*—were significantly downregulated in Wanshu-7.

## 3. Discussion

### 3.1. Effects of Weak-Light Stress on Agronomic Traits and Structural Changes in the Crop

Weak-light stress has a significant impact on the normal growth and development of sweet potatoes. The weakening of light intensity leads to shading reactions that result in tall and slender growth, which promotes an increase in stem length but inhibits diameter [35,36]. The results of our study clearly support this. When the sweet potato was exposed to continuous weak-light stress in the early growth stage, there was a significant increase in vine length and stem diameter, with an increase in vine length and a thinning of the main vine. Even after restoring natural light for half a month, this phenomenon still remained. This finding is consistent with the results reported in [37]. Of the two varieties tested, the long-vine sweet potato variety, Yuhongxinshu-4, was more sensitive to weak-light stress. This is an adaptive adjustment which allows crops to efficiently utilize solar radiation to compensate for the low photosynthesis due to a shaded environment [38].

Weak-light stress also causes leaf thinning, the widening of mesophyll cell gaps, and a decrease in the number of chloroplasts within cells, showing a trend toward a shaded structure [39]. In particular, more than 70% shading makes the plant soft, reducing the ratio of palisade to sponge tissue and increasing the proportion of sponge tissue, which is beneficial for reducing the loss of light quanta in transmitted light and improving light energy utilization efficiency [40,41]. The stronger the shade tolerance, the more leaf thickness decreases, the more sponge tissue develops, and the smaller the palisade sea ratio becomes [42]. In this study, the leaf thickness of Yuhongxinshu-4 decreased significantly compared to Wanshu-7, which further confirms that the former is more tolerant to shade than the latter.

### 3.2. Response of Photosynthetic Abilities of Sweet Potato to Weak-Light Stress

The photosynthetic capacity of plant leaves under weak light is lower than under normal light, and their ability to assimilate CO_2_ decreases [43]. Meanwhile, the net photosynthetic rate rapidly increases with an increase in photosynthetic effective radiation [44]. Light intensity has a significant correlation with the net photosynthetic rate, transpiration rate, stomatal conductance, light saturation point, and light compensation point of leaves. Under weak light, the photosynthetic rate of plant leaves significantly decreases [45]. The results of this study indicate that low-light stress had a significantly inhibitory effect on the net photosynthetic rate of the two different sweet potato varieties. In contrast, intercellular CO_2_ concentration and transpiration rate had increased slightly. This suggests that low-light stress not only reduces the photosynthetic efficiency of plant leaves but also inhibits their ability to absorb carbon. After natural light was restored, the photosynthetic rate was less affected by weak-light stress, indicating that sweet potato leaves can quickly resume normal photosynthesis when returned to a normal light environment.

It has been reported that weak-light stress not only reduces chlorophyll content, but also significantly reduces the activity of the key photosynthetic enzyme rubisco, leading to a decrease in the photosynthetic rate [46,47]. This is somewhat different from this study, in which rubisco activity decreased under low-light conditions, but enzyme activity began to recover after natural light was restored. This is consistent with the changes observed in the photosynthetic rate after full light levels were restored. Conversely, shading increased the chlorophyll content of the sweet potato leaves, with an increase in chlorophyll a and chlorophyll b content but a decrease in the chlorophyll a/b ratio. This is consistent with Zhu’s results [30]. However, another study found that at the growth stage of V12, both chlorophyll a and chlorophyll b significantly decreased after shading treatment [9]. These contradictory findings indicate that the contents of chlorophyll a or b change in different ways for different crops.

### 3.3. The Effect of Periodic Weak Light on Dry Matter Distribution, Yield and Quality

With the weakening of light intensity, the aboveground dry matter weight of sweet potato significantly decreases. Low-light stress first affects the crops’ ability to photosynthesize, changes the plant morphology, then affects dry matter accumulation and distribution, ultimately affecting crop yield and quality [48]. In this study, the aboveground dry matter weight of sweet potatoes decreased significantly during the shading period, although it returned to normal levels after natural light was restored. This indicates that sweet potatoes respond to the environment after light restoration with a significant increase in aboveground dry matter, weakening the adverse effects of early shading. However, the degree of recovery varied among different sweet potato varieties under different light intensities, which may ultimately influence the yield. Weak-light stress not only affects the accumulation of aboveground dry matter weight but also has a certain impact on the development of underground roots. This study found that weak-light stress has a major influence on the development of sweet potato storage roots in the early stage of root expansion. Similar results were also found in maize [49]. This study found that when light levels were 70% lower than natural light, weak-light stress significantly inhibited the root development of two sweet potato varieties, resulting in significant changes in total root length, surface area, and root volume.

Weak light is a stress factor that causes changes in the proportion of crop sources and sinks. The coordination of sources and sinks under weak-light stress reflects the adaptability of crops to environmental stress and alleviates the impact of stress on crop yield [50]. Shading changes the biomass distribution of plants, leading to an increase in the canopy and the ratio of top to root under low-light conditions. This indicates an increase in the proportion of assimilates used to construct photosynthetic organs and support structures, in order to capture as much light energy as possible and maintain carbon assimilation ability, which is a manifestation of plant adaptation to low-light conditions [51]. In this study, the top-to-root ratio of sweet potato increased significantly, which also illustrates the fast growth of vines and leaves but limited storage root expansion. This ratio was significantly higher under weak-light stress than under normal light, indicating that sweet potato biomass was preferentially allocated aboveground in order to adapt to adverse conditions. This is consistent with previous research findings [52].

Insufficient light affects photosynthetic efficiency and carbohydrate accumulation. If protein synthesis remains constant, the yield decreases [53]. This study found that weak-light stress significantly reduced the yield of sweet potato, but the decrease in yield of Yuhongxinshu-4 was less than that of Wanshu-7. The soluble sugar content also declined, but Yuhongxinshu-4 was less affected, reflecting differences between varieties. A similar phenomenon has been observed in tomato fruit [54].

### 3.4. Transcriptome Response of Sweet Potato to Weak-Light Stress

Transcriptomic analysis is a comprehensive study of gene expression in organisms. In recent years, many researchers have focused on comparing gene expression levels in mutant and wild-type organisms under abiotic stresses and normal growth condition, or between different tissues and organs. To date, most studies on the shade tolerance of sweet potatoes have paid attention to physiological and biochemical responses. Transcriptome analysis helps elucidate changes in gene expression related to low-light stress responses and the tolerance mechanism plants use in response to weak-light stress. In our study, the Illumina NovaSeq 6000 sequencing platform was used to perform a transcriptome analysis on two different types of sweet potatoes under low light stress.

The gene expression of the two sweet potato varieties underwent substantial changes under low-light conditions. However, between treatments, there were still many differentially expressed genes after natural light was restored. In particular, the number of genes with altered expression levels under low-light stress was much lower in Yuhongxinshu-4, which is the variety tolerant of weak-light compared to Wanshu-7, which is sensitive to weak-light stress. Furthermore, the number of differentially expressed genes in the two cultivars after natural light was restored was significantly lower than it had been under weak light. A similar phenomenon has been observed with soybean, and the stems of soybean varieties with strong tolerance to weak light levels exhibited more differential transcriptional genes compared to less tolerant varieties [55]. This further illustrates that the two varieties of sweet potato differed in their adaptability to weak light; the variety with greater tolerance to weak light was more stable in terms of its gene expression level. There may be differences in transcriptome changes between leaves and stems of plants under low-light stress.

Through functional annotation and classification of differentially expressed genes, we found that the differentially expressed genes of Yuhongxinshu-4 were enriched in oxidative stress response and response to light stimulation. In contrast, those of Wanshu-7 were enriched in glutathione metabolism and response to light intensity, and two of them had enriched a large number of differentially expressed genes in the oxidoreductase activity under weak-light stress. However, after natural light was restored, it was found that the differentially expressed genes in the two varieties were significantly enriched in photosynthesis, sugar metabolism, chlorophyll metabolism, and response to oxidative stress.

A proteomic analysis of soybeans after shading showed that weak light significantly affects the pathways involved in porphyrin and chlorophyll metabolism, photosynthesis antenna proteins, and photosynthesis [56]. Another study reports that in order to maintain a high rate of photosynthesis in plants under low-light conditions, genes related to photosynthesis and chlorophyll biosynthesis are accelerated in expression [57]. It has also been found that differentially expressed muskmelon genes involved in the glutathione metabolic process of were significantly enriched when plants were shaded [58]. Glutathione can quench reactive oxygen species in cells and is a common antioxidant in plants [59]. These findings suggest that the gene functions of different varieties overlap and also have certain specificities when it comes to responding to weak-light stress. Crops may respond to damage caused by weak-light stress through antioxidant effects. At the same time, according to the enrichment results, it can be inferred that the two sweet potato varieties follow the same metabolic pathway in response to weak-light stress. Nevertheless, both the number of differentially expressed genes involved in each metabolic pathway and the number of up- and down-regulated genes are different, indicating they each have their own specific metabolic pathways which lead to different levels of adaptability to weak-light stress.

## 4. Material and Methods

### 4.1. Sites, Years, and Genotypes

The pot-based experiment was conducted in 2020 and the field experiment was conducted in 2021 at the Institute of Potato Crops (29°49′ N, 106°25′ E), Southwest University, Beibei District, Chongqing, China. The soil in the pot was composed of peat, vermiculite, and sandy loam (volume ratio of 1:1:1). The experimental site is in an area with a subtropical humid monsoon climate with annual rainfall of 1411.4 mm and solar radiation of 2101.8 MJ d^−1^m^−2^.

Our previous study [14], found that Yuhongxinshu-4 is tolerant of weak-light stress, whereas Wanshu-7 is sensitive to it. The two cultivars, bred as the edible type of sweet potato, were selected as experimental materials.

### 4.2. Experimental Setup

The pot experiment in 2020 included weak-light stress treatment and natural light exposure as a control. The two cultivars were transplanted into twelve pots. Weak light (30% transmittance of natural light through a shading net) was chosen as the treatment, and full natural light was used as the control. The treatment was carried out for 30 days post transplanting (DPT), after which the sweet potato seedlings were revealed when the shading net was removed and natural light was restored. The fourth leaves from the top were collected for sequencing samples at 30 DPT and after 15 days post revealing (DPR) after the removal of the shading net. In total, eight different treatments and stages were selected for transcriptome sequencing, and each treatment had three biological replicates.

A completely randomized block design was applied in the field experiment in 2021. The soil in the field was sandy loam with uniform fertility, as shown in Table 6. The weak-light treatment was realized using the same means as in the pot experiment. Full natural light was treated as the control. The weak-light treatment of sweet potato seedlings continued for 60 DPT, after which the shading net was removed and natural light was restored. This simulated the 60-day symbiotic period which occurs when sweet potato and maize are planted together in a relay intercropping system following local production methods.

Each treatment was set with three plots as three replicates. Seedlings were planted in rows with a spacing of 0.62 m between rows and 0.28 m between plants. There were 60 plants in every plot (2 m × 5 m) and the density was 57,000 plants/hm^2^. The photosynthetic characteristics and other related leaf traits were measured at 45 DPT and 15 DPR, and the agronomic traits were measured at the same time. After 150 DPT, the tuberous roots were harvested and the yield of each plot was measured. Water and fertilizer management were based on popular local methods across the entire reproductive period.

### 4.3. Measurements

For each treatment, three sweet potato plants per plot, all in a similar state of growth and status in each plot, were selected for measurements. The fourth and fifth functional leaves from the tops of the plants were put into two separate self-sealing bags. The fourth leaf was stored in an ice box in order to determine its photosynthetic pigment levels and observe its leaf anatomical structure, and the fifth leaf was frozen with liquid nitrogen and stored in a −80 °C ultra-low-temperature refrigerator to determine the rubisco activity. Then, the whole plant was excavated for agronomic traits test.

#### 4.3.1. Stem Diameter

Stem diameter was measured using calipers 10 cm away from the base of the main vine.

#### 4.3.2. Vine Length

Vine length was measured from the base of the sweet potato main vine to the top growth point by means of a band tape.

#### 4.3.3. The Ratio of Shoot to Root (S/R)

After separating the root, stem and leaves of the plant, their fresh weight was measured, and the S/R ratio was calculated.

#### 4.3.4. Aboveground Dry Matter Weight

Fresh stems and leaves were treated at 105 °C for half an hour, dried it at 80 °C to constant weight, and weighed.

#### 4.3.5. Root Morphology

Sweet potato roots were separated into self-sealing bags at 30 DPT, taken back to the laboratory for washing, and analyzed by means of a root scanner (MRS-9600TFU2L, MICROTEK, Delhi, India) to measure their morphological traits.

#### 4.3.6. Yield

The number of plants left and the number of tuberous roots per plant were counted, the tuberous roots were weighed, and the weight was converted to determine the yield per hectare.

#### 4.3.7. The Content of Soluble Protein

This was determined by means of Coomassie Brilliant Blue Staining [60]. A standard curve was prepared using bovine serum protein as a standard solution. A fresh sample weighing 0.5 g was ground into a homogenate, then poured into a 100 mL volumetric flask and allowed to stand for 30 min. The solution was centrifuged at 4000 rpm for 10 min. To 1 mL of the supernatant, 5 mL of Coomassie Brilliant Blue was added for colorimetry at 595 nm using a spectrophotometer. The Calculation formula was:C = (X × V1)/(V2 × M)
where C = the soluble protein content, unit ug/g fresh weight, X = the protein content obtained through standard curves, V1 = total volume of extraction solution, V2 = the volume of extraction solution taken from the sample during measurement, and M = sample quality.

#### 4.3.8. The Content of Soluble Sugar

The soluble sugar content was determined by means of anthrone colorimetry [61]. A standard curve was prepared using sucrose. To 0.2 g of dry sample, 5 mL of water was added. The mixture was boiled in water for 30 min and centrifuged at 4000 rpm for 10 min. The supernatant was extracted twice. Fouty μL of the supernatant was taken and diluted with 50 times the volume of water. To this, 5 mL of 0.2% anthrone sulfuric acid reagent was added and the resultant solution was boiled in water for 1 min. After cooling, a spectrophotometer was used to compare the color at 620 nm. The calculation formula was:Soluble sugar content (%) = [sugar content (μg) × sample volume to volume (mL) × dilution factor]/[reaction solution volume (mL) × sample weight (mg)] × 100

#### 4.3.9. Leaf Anatomical Structure

Fresh leaves were selected to avoid vein sampling (5 mm × 5 mm). The leaves were fixed and preserved with a FAA fixative, dehydrated with ethanol and xylene, and embedded in wax. The thickness of the transverse section was 5 μm. The section was stained with safranine and fast green [62]. A DM500 biological microscope (Leica, Wetzlar, Germany) was used for observation, and an OPcam camera and matched software were used to take photos and measure the thickness of the whole leaf, upper epidermis, palisade tissue, spongy tissue and lower epidermis.

#### 4.3.10. Photosynthetic Gas Parameters

The photosynthetic gas parameters were determined by means of an Li-6400, a portable photosynthesis instrument (LI-COR, Lincoln, NE, USA) [63].

#### 4.3.11. Photosynthetic Pigments

Chlorophyll a and chlorophyll b levels were determined by means of extract solution (consisting of a 4.5 volume of acetone, a 4.5 volume of anhydrous ethanol, and one volume of water), using the digestion extract method [64].

#### 4.3.12. Rubisco Activity

Rubisco activity was measured using a double antibody one-step sandwich enzyme-linked immunosorbent assay [64]. A plant ELISA Kit (Shanghai Youxuan Biotechnology Co., Ltd., Shanghai, China) was used in this test.

#### 4.3.13. Transcriptome Sequencing and Analysis

RNA samples were prepared by Beijing Biomarker Biotechnology Co., Ltd. and sequenced by the Illumina NovaSeq 6000 sequencing platform. The FPKM method [65] was used to screen differentially expressed genes and calculate the gene expression levels. DESseq2 (https://bioconductor.org/packages/release/bioc/html/DESeq2.html (accessed on15 January 2021)) was used to analyze differential expression between samples. During the detection of differentially expressed genes, the fold change (FC) ≥ 2 and the false discovery rate (FDR) < 0.01 were used as screening criteria. The FC refers to the ratio of expression between two groups of samples, and the *p*-value represents the significant difference in the corrected FDR. The differentially expressed genes were compared with the protein sequences in the gene ontology (GO) database to obtain the protein function annotation and classification statistics corresponding to the differentially expressed genes.

### 4.4. Statistical Analysis

Microsoft Excel 2016 was used for data processing, DPS 7.05 was used for analysis of variance and LSD multiple comparison was conducted, and GraphPad Prism 8 was used for drawing.

## 5. Conclusions

This study found that, under weak light conditions, the sweet potato stems became slenderer, leaves became thinner, and root development was inhibited. Furthermore, weak-light stress inhibited the activity of the key photosynthetic enzyme rubisco and reduced the photosynthetic rate. However, by increasing chlorophyll b, diffuse light was more easily absorbed, and weak-light stress was decreased. After natural light was restored, some physiological and biochemical effects gradually approached normal levels, but the yield ultimately decreased. The two varieties with different sensitivities to weak light share metabolic pathways and have characteristic metabolic pathways, and there were differences in the number of differentially expressed genes involved in various metabolic pathways and the number of up-regulated and down-regulated genes. These results provide a good explanation of the fundamental reasons for the decline in the yield of sweet potatoes under weak light conditions in terms of morphological traits, physiological levels and expression of some key genes. Since this experiment was not carried out in a relay intercropping system of maize/sweet potato but in simulated weak light, we intend to further test these findings in a real system in order to identify the most appropriate configuration for relay intercropping in order to obtain the best arrangement for sweet potato production.

## Figures and Tables

**Figure 1 plants-13-02214-f001:**
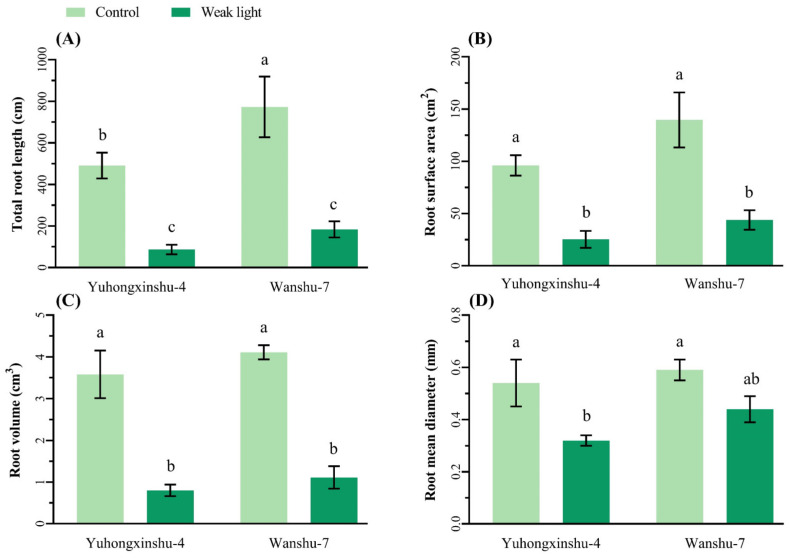
Effects of weak-light stress on root morphology of different sweet potato cultivars. The column height represents the LSD value at each sampling date, and different letters indicate the significance at *p* < 0.05. (**A**) total root length, (**B**) root surface, (**C**) root volume, (**D**) root mean diameter.

**Figure 2 plants-13-02214-f002:**
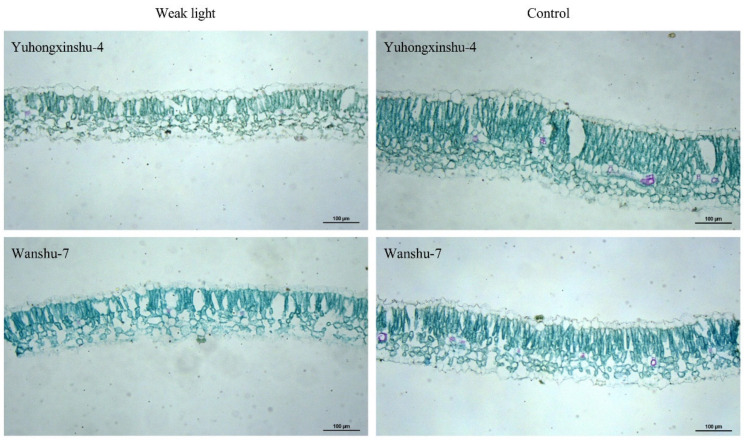
Anatomy of a sweet potato leaf after 45 days of shading.

**Figure 3 plants-13-02214-f003:**
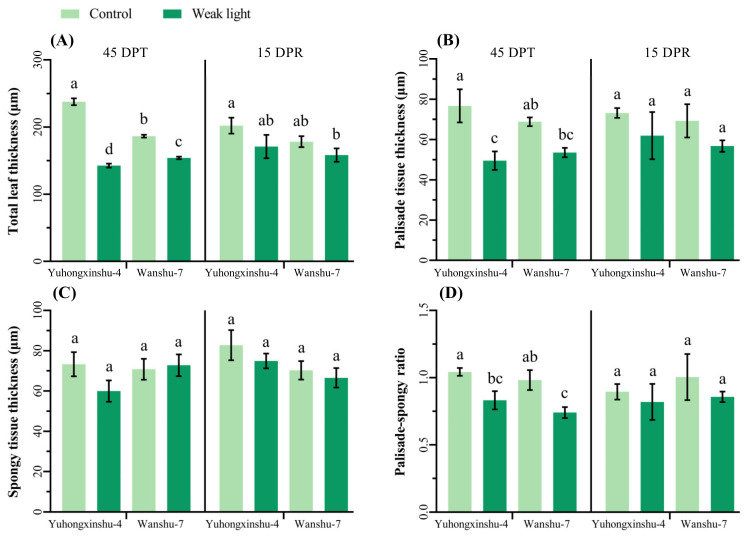
Effects of weak light stress on leaf anatomical structure of different sweet potato cultivars. The columns height represents the LSD value at each sampling date, and different letters indicate the significance at *p* < 0.05. (**A**) leaf thickness, (**B**) palisade tissue thickness, (**C**) palisade tissue thickness, (**D**) ratio of palisade tissue to sponge tissue.

**Figure 4 plants-13-02214-f004:**
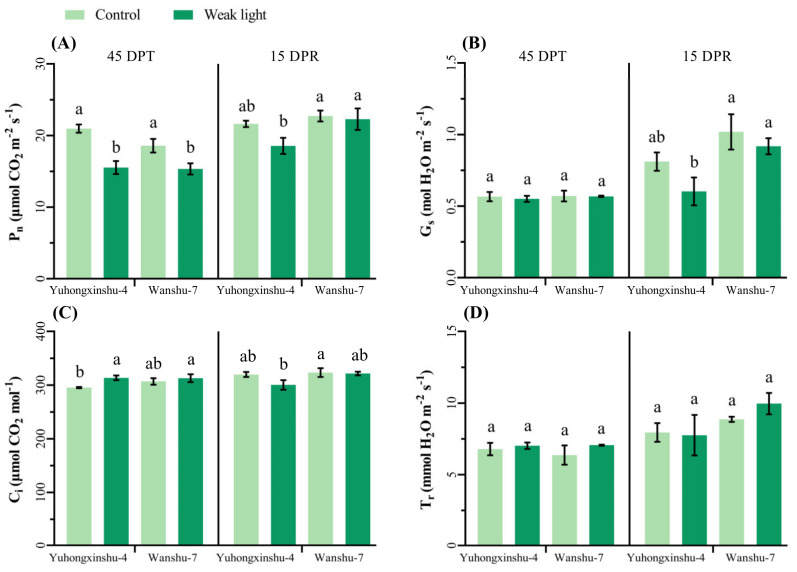
Effects of weak-light stress on photosynthetic gas exchange parameters of different sweet potato cultivars. The column height represents the LSD value at each sampling date, and different letters indicate the significance at *p* < 0.05. (**A**–**D**) represent Pn, Gs, Ci and Tr, respectively.

**Figure 5 plants-13-02214-f005:**
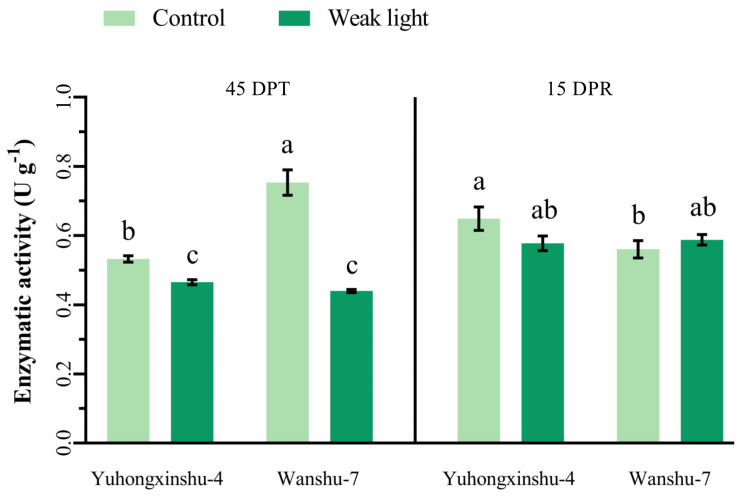
Effects of weak-light stress on the activity of rubisco enzymes in leaves of different sweet potato cultivars. The column height represents the LSD value at each sampling date, and different letters indicate the significance at *p* < 0.05.

**Figure 6 plants-13-02214-f006:**
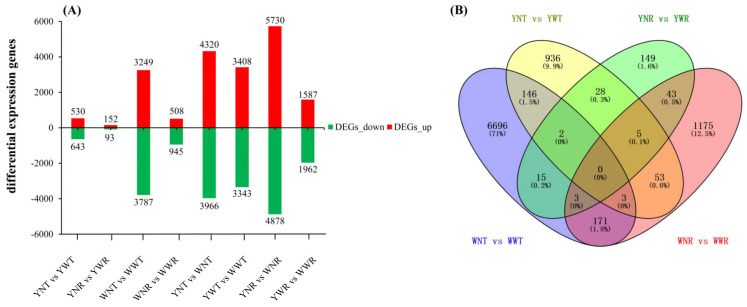
The number of differentially expressed genes in different treatments. YNT: leaf sample of Yuhongxinshu-4 in natural light at 30 DPT; YWT: leaf sample of Yuhongxinshu-4 in weak light at 30 DPT; YNR: leaf sample of Yuhongxinshu-4 in natural light at 15 DPR; YWR: leaf sample of Yuhongxinshu-4 in weak light at 15 DPR; WNT: leaf sample of Wanshu-7 in natural light at 30 DPT; WWT: leaf sample of Wanshu-7 in weak light at 30 DPT; WNR: leaf sample of Wanshu-7 in natural light at 15 DPR; WWR: leaf sample of Wanshu-7 in weak light at 15 DPR. (**A**) The number of differentially expressed genes for every treatment, (**B**) The number of differentially expressed genes co-expressed.

**Figure 7 plants-13-02214-f007:**
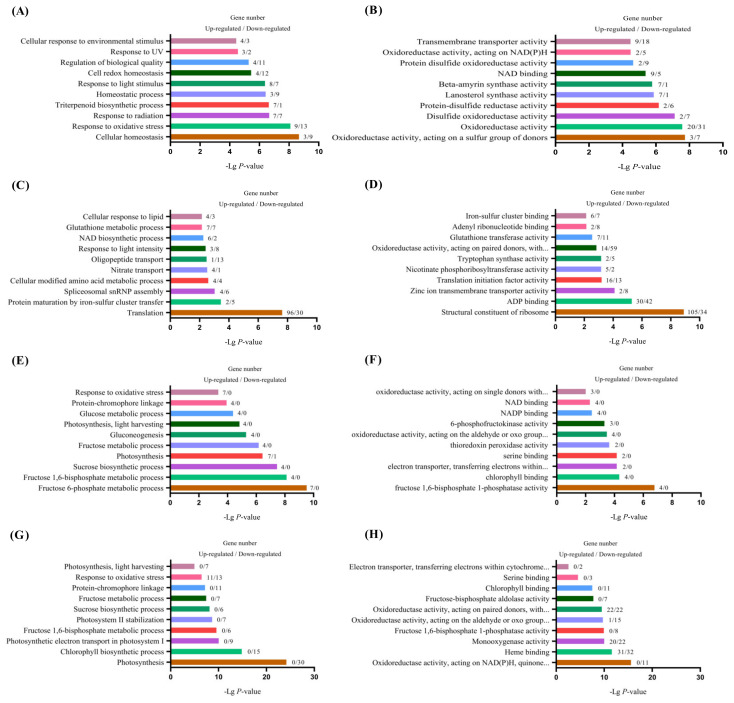
GO enrichment of differentially expressed genes under weak-light stress. Shading period data: (**A**) GO biological process of Yuhongxinshu-4, (**B**) GO molecular function of Yuhongxinshu-4, (**C**) GO biological process of Wanshu-7, (**D**) GO molecular function of Wanshu-7. Restore data after lighting: (**E**) GO biological process of Yuhongxinshu-4, (**F**) GO molecular function of Yuhongxinshu-4, (**G**) GO biological process of Wanshu-7, (**H**) GO molecular function of Wanshu-7.

**Table 1 plants-13-02214-t001:** Effects of weak-light stress on the agronomic characters of different sweet potato cultivars.

Time	Cultivars	Treatments	Stem Diameter (cm)	Vine Length (cm)	Shoot Dry Matter (g plant^−1^)	S/R
45 DPT	Yuhongxinshu-4	Control	0.70 ± 0.05 ab	140.00 ± 12.77 b	29.28 ± 2.54 a	26.57 ± 6.97 bc
		Weak light	0.55 ± 0.02 c	240.24 ± 20.69 a	18.69 ± 0.58 b	47.35 ± 4.76 a
	Wanshu-7	Control	0.79 ± 0.04 a	92.32 ± 5.67 c	28.53 ± 1.53 a	17.98 ± 2.49 c
		Weak light	0.63 ± 0.03 bc	125.22 ± 10.16 bc	19.86 ± 2.66 b	34.20 ± 1.06 ab
15 DPR	Yuhongxinshu-4	Control	0.58 ± 0.04 b	223.56 ± 24.36 b	64.53 ± 2.73 a	9.52 ± 2.32 b
		Weak light	0.51 ± 0.02 b	410.11 ± 26.16 a	62.30 ± 7.73 a	41.84 ± 6.35 a
	Wanshu-7	Control	0.70 ± 0.03 a	194.89 ± 13.20 b	57.52 ± 5.00 a	4.20 ± 1.10 b
		Weak light	0.56 ± 0.02 b	204.44 ± 18.06 b	49.96 ± 4.53 a	39.85 ± 8.53 a

S/R is the weight ratio of aboveground parts (including stems and leaves) to underground parts (including storage roots and other roots). Values are the means of 3 replicates. Similar letters in a column at the same time indicate no significant difference by LSD-test (α = 0.05). This is the same as below.

**Table 2 plants-13-02214-t002:** Effects of weak-light stress on yield and quality of different sweet potato cultivars.

Cultivars	Treatments	Number of Storage Root	Storage Root Yield (kg·hm^−2^)	Soluble Protein Content (mg g^−1^)	Soluble Sugar Content (%)
Yuhongxinshu-4	Control	1.52 ± 0.18 a	32,408.38 ± 2011.92 a	7.91 ± 0.49 ab	22.15 ± 2.46 ab
	Weak light	0.89 ± 0.20 b	20,898.96 ± 212.44 b	8.61 ± 0.35 a	19.33 ± 0.78 b
Wanshu-7	Control	1.96 ± 0.15 a	31,361.43 ± 1789.20 a	7.04 ± 0.58 b	26.85 ± 1.41 a
	Weak light	1.46 ± 0.16 a	18,239.09 ± 2272.24 b	7.91 ± 0.26 ab	19.13 ± 1.61 b

Different letters behind the numbers in the same column indicate the significance at *p* < 0.05.

**Table 3 plants-13-02214-t003:** Effects of weak-light stress on chlorophyll content in leaves of different sweet potato cultivars.

Time	Cultivars	Treatments	Chlorophyll a	Chlorophyll b	Chlorophyll a/b
45 DPT	Yuhongxinshu-4	Control	1.07 ± 0.04 c	0.29 ± 0.01 c	3.67 ± 0.05 c
		Weak light	1.42 ± 0.07 b	0.42 ± 0.01 a	3.37 ± 0.11 c
	Wanshu-7	Control	1.29 ± 0.12 bc	0.26 ± 0.02 c	4.95 ± 0.21 a
		Weak light	1.66 ± 0.02 a	0.38 ± 0.01 b	4.37 ± 0.07 b
15 DPR	Yuhongxinshu-4	Control	1.22 ± 0.06 a	0.31 ± 0.02 a	3.93 ± 0.11 a
		Weak light	1.14 ± 0.04 a	0.31 ± 0.01 a	3.7 ± 0.08 a
	Wanshu-7	Control	1.25 ± 0.12 a	0.33 ± 0.03 a	3.77 ± 0.20 a
		Weak light	1.32 ± 0.09 a	0.33 ± 0.02 a	3.98 ± 0.01 a

Different letters behind the numbers in the same column indicate the significance at *p* < 0.05.

**Table 4 plants-13-02214-t004:** Transcriptome sequencing data and statistics of comparison with reference genome.

Classification	Maximum	Minimum	Average
Total reads	53,027,458	38,556,254	44,006,063
Mapped reads	40,757,234	28,780,139	33,206,125
Ratio of mapped reads (%)	77.22	71.66	75.43
GC content (%)	46.48	43.21	45.24
Percentage of Q30base (%)	94.59	93.54	94.01

**Table 5 plants-13-02214-t005:** Some differential genes related to weak-light stress response.

Comparison Group	Gene Number	Log_2_ (FC)	Nr Annotation
YNT vs. YWT	Tai6.12471	−1.080901851	malate dehydrogenase, glyoxysomal-like
	Tai6.44472	−1.603626351	chloroplast envelope quinone oxidoreductase homolog
	Tai6.21350	−1.914736278	ferredoxin—NADP reductase, leaf-type isozyme, chloroplastic-like
	Tai6.18802	−1.987373714	photosystem II 22 kDa protein, chloroplastic
	Tai6.31070	−2.37509445	flavonol synthase/flavanone 3-hydroxylase-like
WNT vs. WWT	Tai6.42466	4.164479306	ferredoxin, root R-B2-like
	Tai6.15090	2.123567865	glyceraldehyde-3-phosphate dehydrogenase 3, cytosolic
	Tai6.44472	−1.885637254	chloroplast envelope quinone oxidoreductase homolog
	Tai6.31070	−2.489474572	flavonol synthase/flavanone 3-hydroxylase-like
YNR vs. YWR	Tai6.28593	2.086727777	photosystem II 22 kDa protein, chloroplastic
	Tai6.19673	2.151086707	photosystem II 22 kDa protein, chloroplastic
	Tai6.18802	1.896023004	photosystem II 22 kDa protein, chloroplastic
WNR vs. WWR	Tai6.21350	−2.071930587	ferredoxin—NADP reductase, leaf-type isozyme, chloroplastic-like
	Tai6.35389	−2.26692449	glyceraldehyde-3-phosphate dehydrogenase B, chloroplastic
	Tai6.27478	−1.478949039	psbP domain-containing protein 4, chloroplastic isoform X1
	Tai6.12471	−1.617623562	malate dehydrogenase, glyoxysomal-like

**Table 6 plants-13-02214-t006:** The basic soil fertility in the field experiment.

pH	Organic Matter (g kg^−1^)	Total Nitrogen (g kg^−1^)	Total Phosphorus (g kg^−1^)	Total Potassium (g kg^−1^)	Alkali Hydrolyzable Nitrogen (mg kg^−1^)	Available Potassium (mg kg^−1^)	Available Phosphorus (mg kg^−1^)
7.45	21.44	0.83	1.03	18.77	92.81	101.22	18.23

## Data Availability

The datasets generated during and/or analyzed during the current study are available from the corresponding author upon reasonable request.

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
