# Peer review of "Physiological and Transcriptome Responses of Sweet Potato [Ipomoea batatas (L.) Lam] to Weak-Light Stress"

_plants, 2024, doi:10.3390/plants13162214_

Round 1

Reviewer 1 Report

Comments and Suggestions for Authors

Dear Authors,

my comments on the manuscript are as follows: the manuscript provides the basis for breeding varieties of sweet potato with low light tolerance. The topic is a good one, as we need everything that can be used as a topic due to the growing global population. Sweet potato is an important food crop and its breeding is worthwhile.

I suggest the Abstract should be no more than 200 words. Therefore, it would be worth rewriting the current one and shortening the text, clarifying both the objective and the results.

Key words are appropriate.

Introduction: basically well structured, but many old publication results. It would be useful to significantly expand the chapter with publication results from the last 5 years. 

In parallel, it would be useful to structure the Discussion chapter - it would be useful to look for deeper connections with the more recent results discussed in the Introduction chapter. I propose to amend the chapter on this basis. 

4.3. should not be in bold, suggest using a different notation. 

The figures are of very high quality. 

Author Response

Comment 1: the manuscript provides the basis for breeding varieties of sweet potato with low light tolerance. The topic is a good one, as we need everything that can be used as a topic due to the growing global population. Sweet potato is an important food crop and its breeding is worthwhile. (Thank you!)

Comment 2: I suggest the Abstract should be no more than 200 words. Therefore, it would be worth rewriting the current one and shortening the text, clarifying both the objective and the results.

Response: We rewrote the abstract with less words from 320 to 270. (page1, line 13-32)

Comment 3: Key words are appropriate. (Thank you!)

Comment 4: basically well structured, but many old publication results. It would be useful to significantly expand the chapter with publication results from the last 5 years.

Response: We updated and cited some latest literatures, such as [2], [3] and [9]. (page 17, line 622-627; Page 18, line 638-639)

Comment 5: In parallel, it would be useful to structure the Discussion chapter - it would be useful to look for deeper connections with the more recent results discussed in the Introduction chapter. I propose to amend the chapter on this basis. 

Response: We revised that part and cited a new report [9] in Discussion chapter. (page 11-14, line 312-448, in particular line 357-360)

Comment 6: 4.3. should not be in bold, suggest using a different notation. 

Response: We revised them according to the regulation of this journal. (page15-16, line 494-567)

Comment 7: The figures are of very high quality. (Thank you!)

Reviewer 2 Report

Comments and Suggestions for Authors

This manuscript by Yang et al. found that Under low light stress, the anatomical structure of functional leaves of sweetpotato changed significantly, the activity of photosynthetic enzyme Rubisco decreased, and the net photosynthetic rate decreased. By KEGG analysis, the differentially expressed genes in two different sweetpotato varieties under low light stress had the same enrichment pathway, mainly in glutathione metabolism and flavonoid biosynthesis. However, after restoration of light, the differentially expressed genes were enriched in photosynthesis, photosynthetic pigment synthesis and carbon metabolism pathways. With these results, the authors demonstrated that low light was able to alter sweetpotato plant morphology, reduce plant dry matter weight, affect root development, photosynthetic physiology and expressed gene levels, and ultimately lead to tuber yield loss. The amount of work in this manuscript is impressive and figures are of reasonable quality, but the authors need to clarify and improve a little points.

Q1: There are a few Chinglish phenomena in the text. The language should be carefully edited by a native English speaker with strong background in plant science.

Q2:Your abstract is too long. You should simplify it to make it easier for readers to understand the overall content of your article.

Q3:Please check carefully and modify the format of the references, there are many errors. And the format is not standardised.

Q4: For figure 3A,B and 4A,B, please add a horizontal axis.

Q5: There are too many tables in your main text. Please put them in the appendix.

Comments on the Quality of English Language

 Extensive editing of English language required.

Author Response

Q1: There are a few Chinglish phenomena in the text. The language should be carefully edited by a native English speaker with strong background in plant science.

Response: The revised version was edited in English by MDPI. (English-Editing-Certificate-83402)

Q2:Your abstract is too long. You should simplify it to make it easier for readers to understand the overall content of your article.

Response: We rewrote the abstract with 270 words less than previous one. (page1, line 13-32)

Q3:Please check carefully and modify the format of the references, there are many errors. And the format is not standardised.

Response: We checked references one by one and revised those errors. (page 17-20, line 618-759)

Q4: For figure 3A,B and 4A,B, please add a horizontal axis.

Response: We Added the horizontal axis on those figures and substituted these figures with high quality. (page 6, Figure 3; page 7, Figure 4)

Q5: There are too many tables in your main text. Please put them in the appendix.

Response: This is out of question, but it may be determined by editor and we haven’t put them in the appendix in this version temporarily.

Reviewer 3 Report

Comments and Suggestions for Authors

The manuscript entitled "Physiological and transcriptome responses of sweet potato to weak light stress’’ investigated the reaction of two sweet potato cultivars to light stress, clarified the changes and differences in agronomic and physiological traits between weak and normal light, and examined the divergence of gene expression related to light and photosynthesis’’. The structure of the paper is well, the idea of the research is interesting, and the authors provided interesting results and clear methods. Overall, the paper is clear about what the authors are looking for. There are only a few comments, please see them as below:

·         The abstract is too long! The structure of the abstract could be better, the framework of the abstract section includes a brief introduction, material and method, results, and an impressive conclusion. The authors should mention the materials and methods briefly.

·         Also, keywords should be different from the title, make them specific. I would suggest that add the scientific name of sweet potato as one of the keywords.

·         Line 43: unclear sentence, paraphrase!

·         I would suggest that the authors add some previous research about the effect of light stress on sweet potatoes and maize, compare it to the current study, and highlight the gap of knowledge in paragraph two! In your paper, there isn’t any connection between current research and previous ones.

·         Figures 1, 3,4: The resolution of the axis’s titles is very low! The Cultivars are unreadable. Replace them with high-resolution ones.

·         Replace figures 6 and 7 with high-resolution ones.

·         Table 1: add the definition of the abbreviation S/R in the table description.

·         The main problem of the discussion section is the lack of a good connection between the current research and previous ones and in many cases, it’s the repetition of the results.

·         Section 4.1. Sites, years, and genotypes -- > I would suggest that the authors provide soil characteristics in the form of a Table in the different section.

·         Section 4.3. Measurements  -- > Add references for each one!

·         Add the suggestion for future research and research limitations at the end of the conclusion section.

·         There are not any 2023-2024 references in the references list. Update this section!

Comments on the Quality of English Language

Minor editing of English language required

Author Response

Comment 1: The abstract is too long! The structure of the abstract could be better, the framework of the abstract section includes a brief introduction, material and method, results, and an impressive conclusion. The authors should mention the materials and methods briefly.

Response: In revised version, we complemented the material and method in the abstract, and shortened it from 320 to 270 words. (page1, line 13-32)

Comment 2:  Also, keywords should be different from the title, make them specific. I would suggest that add the scientific name of sweet potato as one of the keywords.

Response: We substituted sweet potato with ‘Ipomoea batatas (L.) Lam’. (page1, line 33)

Comment 3:  Line 43: unclear sentence, paraphrase! (Revised it. Page 1, line 39-43)

Comment 4:  I would suggest that the authors add some previous research about the effect of light stress on sweet potatoes and maize, compare it to the current study, and highlight the gap of knowledge in paragraph two! In your paper, there isn’t any connection between current research and previous ones.

Response: In paragraph two, we only complemented one report [14] about the effect of weak light stress on sweet potatoes, which is highly relevant to our topic. Because maize is the pre crop in the relay intercropping system, it is affected little due to sweet potato in the production system. Only a few researchers pay attention to relay intercropping maize and sweet potato in China, and there are few reports published about that.

Comment 5:  Figures 1, 3, 4: The resolution of the axis’s titles is very low! The Cultivars are unreadable. Replace them with high-resolution ones.  Replace figures 6 and 7 with high-resolution ones.

Response: We substituted these figures with high-quality ones. And please check figure 1, 3, 4, 6, 7 in revised version.

Comment 6: Table 1: add the definition of the abbreviation S/R in the table description.

Response: We added the definition of the abbreviation below Table 1. (page 4, line 138-139)

Comment 7:  The main problem of the discussion section is the lack of a good connection between the current research and previous ones and in many cases, it’s the repetition of the results.

Response: We revised that section and complemented a few more appropriate and relational literatures. (page 11-14, line 312-448)

Comment 8: Section 4.1. Sites, years, and genotypes -- > I would suggest that the authors provide soil characteristics in the form of a Table in the different section.

Response: The soil characteristics was shown in table 6 in Section 4.2. (page 14, line 478)

Comment 9:    Section 4.3. Measurements  -- > Add references for each one(Added [62,63,64], page16, line 548, 555, 560, 564) ! 

Comment 10: Add the suggestion for future research and research limitations at the end of the conclusion section

Response: We added them in that section. (page 17, line 596-600)

Comment 11:  There are not any 2023-2024 references in the references list. Update this section!

Response: We complemented some latest literatures, such as [2], [3], [9] in new version, and deleted [7] in old version.

Round 2

Reviewer 1 Report

Comments and Suggestions for Authors

I recommend it for publication